# First Evidence of the Presence of the Causative Agent of Caseous Lymphadenitis—*Corynebacterium pseudotuberculosis* in Dairy Products Produced from the Milk of Small Ruminants

**DOI:** 10.3390/pathogens11121425

**Published:** 2022-11-26

**Authors:** Denisa Langova, Iva Slana, Jana Okunkova, Monika Moravkova, Martina Florianova, Jirina Markova

**Affiliations:** Department of Microbiology and Antimicrobial Resistance, Veterinary Research Institute, 62100 Brno, Czech Republic

**Keywords:** food safety, zoonosis, small ruminants, dairy products, CLA, paratuberculosis, MRSA, qPCR, cultivation

## Abstract

This study focused on the detection and quantification of selected bacteria and on the presence of enterotoxin genes in milk and dairy products from sheep and goat farms in the Czech Republic using quantitative real-time PCR (qPCR) and multiplex PCR (PCR). The presence of *Corynebacterium pseudotuberculosis* (CP), *Mycobacterium avium* subsp. *paratuberculosis* (MAP), *Listeria monocytogenes*, *Staphylococcus aureus*, *S. aureus* enterotoxin genes and methicillin-resistant *Staphylococcus aureus* (MRSA) was determined in 18 milk samples, 28 fresh cheeses, 20 ripened cheeses and 14 yoghurts. The serological status of the herds in relation to CP and MAP was taken into account. The most frequently detected bacterium was *S. aureus* (48.8%), and subsequent PCR revealed 11 MRSA positive samples. The *S. aureus* enterotoxin genes *seg*, *sei* and *sec* were detected in two goat cheeses. Cheese samples showed a statistically higher risk of SA and MRSA occurrence. CP (8.8%) and MAP (13.8%) were detected by qPCR on two different seropositive farms. Cultivation of qPCR positive CP samples on agar plates supplemented with potassium tellurite showed the presence of viable bacterium. The results obtained confirmed the necessity of monitoring the infectious status of dairy animals and rapid diagnosis of bacterial pathogens in milk and dairy products.

## 1. Introduction

Goat and sheep’s milk and dairy products from small craft farms are a popular alternative to dairy products made from cow’s milk in the Czech Republic. However, raw milk is rich in the presence of various groups of microorganisms, which may include pathogenic species and these may pose a risk to consumers due to the transmission of food-borne diseases, particularly in relation to the consumption of unpasteurized milk and dairy products [1]. According to the European Food Safety Authority (EFSA), bacteria (e.g., *Listeria monocytogenes*, *Salmonella* spp., *Mycobacterium* spp.) and bacterial toxins (e.g., *Staphylococcus aureus* toxins, shigatoxin produced by *Escherichia coli*) are common causative agents of alimentary diseases. These diseases are also frequently associated with the consumption of contaminated milk and dairy products and contamination may occur during all stages of the food chain through, for example, infectious processes in animals such as mastitis, or secondary to milking and subsequent milk processing [1,2].

The major causative agent of intramammary infections in small ruminants is *Staphylococcus aureus* (SA), which causes economic losses for livestock farmers [3,4]. Various enterotoxins produced as metabolic products of SA can cause alimentary intoxications [5,6]. While the bacteria are inactivated by pasteurization, the toxins remain active even after heat treatment of the milk. In some cases, SA infection is difficult to control due to its ability to develop resistance to almost all classes of antimicrobial agents, especially β-lactam antibiotics (known as methicillin resistance) [7]. The expression of a PBP2a (or PBP2’) protein encoded by the *mecA* gene and its significantly lower ability to bind to the antibiotic is a major cause of these resistances [8]. 

Pathogenic bacterial species of the genus *Corynebacterium* may also be present in ruminant milk [9]. This genus consists of a heterogeneous group of Gram-positive bacteria including many species that have been isolated from human and veterinary infections such as cystitis, mastitis, hyperkeratosis, etc., [10]. Some corynebacteria belong to a potentially toxigenic group and are associated with the production of exotoxins, which are important molecules in terms of bacterial virulence. This group also includes the intracellular bacterium *Corynebacterium pseudotuberculosis* (CP), the causative agent of caseous lymphadenitis (CLA) [10]. CLA is a chronic disease and affects flocks of small ruminants all over the world [11]. Infection is associated with abscess formation, which may occur in two forms, superficial and visceral [12]. Some cases of CP spreading through the milk of an infected animal have been reported when an abscess has occurred in the mammary gland [13,14] or as a member of mixed infections in the course of subclinical mastitis [15]. However, the occurrence of non-pathogenic corynebacteria in raw ruminant milk has also been described [9,16]. Over the years, the development of multi-drug resistance in some species of the genus *Corynebacterium* has been demonstrated, which brings emphasis to the search for new therapeutic approaches in the treatment of infections [17].

Paratuberculosis (PTB) is another significant infectious disease of small ruminants in the context of food safety. PTB caused by *Mycobacterium avium* subs. *paratuberculosis* (MAP) is widespread and poses a risk to sheep and goat farming because it tends to remain hidden, often showing only indirect effects on production [18]. To date, MAP has been associated with Crohn’s disease, sarcoidosis and Blau syndrome in humans, for which reason the shedding of bacteria into the milk of affected animals may pose a serious health risk to consumers [19].

Compared to other foodborne pathogens, *Listeria monocytogenes* (LM) probably has a lower pathogenic potential for humans due to the high infectious dose, but the level of host susceptibility must always be taken into account. Certain species of mammals (e.g., sheep) are more susceptible to infection, and the bacterium may also be excreted for a long time even in the milk of animals without clinical signs [20,21].

The processing of dairy products on small sheep and goat farms and their direct sale to consumers is expanding in the Czech Republic. Few studies have focused on detection of this pathogen in milk and milk products from CLA-affected herds considering the widespread distribution of CP in goats and sheep. The aim of this study was, therefore, to detect and quantify CP along with MAP in sheep and goat’s milk and dairy products in the Czech Republic taking into account the serological status of the animals. Detection was also extended to include SA, LM, genes responsible for SA enterotoxins and the presence of methicillin-resistant *S. aureus* (MRSA) in the context of food safety.

## 2. Results

### 2.1. Presence of C. pseudotuberculosis and M. avium subsp. paratuberculosis in Tested Samples

The presence of DNA from *C. pseudotuberculosis* (CP) was revealed in milk and dairy products on one from 12 farms. This was a pseudotuberculosis seropositive farm with dairy breeds of sheep and goats (around 300 animals), that sold milk and dairy products directly on the farm and to the market. CP was detected in one fresh and two ripened goat’s milk cheeses, in unpasteurized sheep’s milk and in three fresh sheep’s milk cheeses. Quantification of CP per gram of cheese or mL of milk ranged from 10^0^ to 10^1^ (Table 1). Further, CP qPCR positive samples were cultured on agar plates supplemented with and without potassium tellurite (K₂TeO₃). Strain viability was determined based on the positivity of qPCR lysates prepared from mixed bacterial cultures grown from one milk sample and three cheese samples on agar plates supplemented with potassium tellurite. To the author’s knowledge, this is the first finding of viable CP in the milk products of small ruminants. The bacterium was not detected on the other tested farms, even in those with similar seroprevalence.

Similarly, MAP DNA in milk and dairy products was detected by qPCR on only one of the twelve tested farms. The positive samples originated from a PTB seropositive farm with dairy breeds of sheep and goats (around 1500 animals), which also supplied dairy products to the market. MAP positivity was found in tested samples from both sheep and goats. The number of MAP genome equivalent ranged from 10^0^ to 10^3^ per gram or mL of sample (Table 1). Dairy products and milk were MAP negative on other farms, both serologically negative and positive for PTB.

The results of serological screening for the presence of antibodies to CP and MAP on the tested farms along with the results of detection and quantification of zoonotic pathogens are presented in Table 2.

### 2.2. Detection of S. aureus and L. monocytogenes

*S. aureus* (SA) was the most frequently detected bacterium in the milk and dairy products. qPCR detected SA on nine of the twelve tested farms from these samples. A total of 39 of 80 tested samples (48.8%) were qPCR positive, including eight milk samples, fifteen fresh cheeses, fourteen ripened cheeses and two yoghurts (Table 1). In the majority of cases, SA was detected in both milk samples and dairy products on the same farm. The only exceptions were two farms that had positive dairy products but negative milk samples. There was no association with SA positivity by type of farm or production size. The quantification of SA in milk and dairy products ranged from 10^0^ to 10^5^ genome equivalent per one mL or one g of tested sample and positivity was recorded in both qPCR targets (SA442, *nuc*). All samples were qPCR negative for the presence of *Listeria* species (spp.) and *L. monocytogenes*.

The conventional multiplex PCR (PCR) method for the identification of SA and methicillin-resistant *S. aureus* (MRSA) detected SA DNA in 27 samples (39.7%). MRSA was identified in 11 of these positive samples (six fresh goat cheeses, three ripened sheep cheeses and two ripened goat cheeses) from five different farms. In most cases, PCR detected SA in SA qPCR positive samples with a number of genome equivalent ≥10^2^ per one g or one mL. Genes for staphylococcal enterotoxins were found in two samples of fresh goat cheese from two different farms. Specifically, these were *seg* and *sei* genes for enterotoxins G and I, respectively, in one cheese and the *sec* gene for enterotoxin C in cheese from a second farm. Both farms produced dairy products only for their own consumption and the number of genome equivalent after SA qPCR for these two cheese samples were 10^5^ per gram of product.

### 2.3. Statistical Analysis

An association between the three categories of tested samples (milk, cheeses and yoghurts) and the frequency of positivity of the bacteria in these individual categories was only demonstrated for the results of SA qPCR (*p* < 0.01; Fisher’s exact test) and MRSA detection (*p* < 0.05; Fisher’s exact test). In SA qPCR results, pairwise testing showed a statistically significant difference between cheeses and yoghurts (*p* < 0.01; Fisher’s exact test). A total of 29/48 (60.4%) cheeses were SA positive, while only 2/14 (14.3%) yoghurts were positive. The odds ratio (OR) was 9.16 (95% CI: 2.11–43.26). Based on these data, the odds of SA-positivity were approximately nine times higher for cheese samples than for yoghurts. In the case of MRSA, pairwise testing showed a statistically significant difference between milk and cheeses (*p* < 0.05; Fisher’s exact test). A total of 11/48 (22.9%) of the cheeses were positive for MRSA, while no MRSA was detected in the milk samples. It was not possible to reliably estimate the OR due to zero findings of MRSA in milk. Of all three examined categories of samples, cheeses showed a statistically higher risk of SA and MRSA occurrence.

The prevalence of SA on the MAP antibody-positive farms was statistically significantly higher than on the MAP negative farms (*p* < 0.05; Fisher’s exact test). The OR was 3.32 (95% CI: 1.15–9.74), and samples from MAP positive farms therefore had a more than 3 times higher chance of SA-positivity than samples originating from MAP negative farms. The prevalence of SA in the CP antibody-positive group of farms was 31.37% (16/51), whereas it was 85.71% (24/28) in the CP dubious group. The difference was statistically significant (*p* < 0.01; Fisher’s exact test). The OR was 13.13 (95% CI: 3.84–28.26), and the odds of SA-positivity were therefore around 13 times higher in the CP dubious group than in the CP positive group of farms.

## 3. Discussion

The presence of *C. pseudotuberculosis* (CP) in milk and dairy products was recorded on only one farm in our study with a prevalence of 8.8% of all 80 tested samples. The history of caseous lymphadenitis (CLA) infection on this farm dates back to 2015 and the overall CP seroprevalence in animals was 68.2% (89.5% in goats and 45.3% in sheep) which was the highest value of all CLA positive farms. Other high seroprevalences were measured on one goat (44.7%) and one sheep (34.7%) farm, but their samples were negative for CP. Since, according to the available data in the literature, CP has not yet been isolated from sheep and goat dairy products, qPCR positive samples were cultured to determine the viability of the bacterium. A selective medium containing tellurite is essential for primary isolation of Corynebacteria. However, dairy products are naturally contaminated by large numbers of tellurite-resistant Gram-positive bacteria, which has complicated CP isolation. Therefore, the qPCR method was used to examine swabs from agar plates containing mixed cultures of tellurite-tolerant microorganisms. Positive results from the examination of these bacterial lysates indicate the viability of the bacterium in the product because the process of sample preparation, subsequent dilution and use of the culture method virtually eliminates the possibility of detecting residual DNA from the original qPCR positive samples that ranged from 10^0^ to 10^1^ of genome equivalent per mL or g. Since excretion of CP into milk has been recorded when an abscess occurs directly in the mammary gland [13,14], and assuming proper inspection of the udder before each milking, contamination could have been caused by inadequate hygiene during milking, processing and product storage. However, we cannot exclude the possibility of subclinical mastitis in dairy animals [15]. Although human disease after consumption of CP contaminated milk is not frequently reported [24] and cases of infection from dairy products are not mentioned in the literature, these sources of infection should still be considered at risk. Especially in the case of viable bacteria and the level of susceptibility of the organism which the CP enters with the contaminated dairy product.

Similarly to CP, IS*900* qPCR revealed *Mycobacterium avium* subs. *paratuberculosis* (MAP) in milk and dairy products on only one farm. This farm had the highest MAP seroprevalence (16.6%) of all the tested farms. These results were further confirmed and more precisely quantified by qPCR targeting *F57*. The use of two MAP qPCR detection systems ensured both highly sensitive qualitative detection in the case of multi-copy IS*900* and accurate quantitative detection with single-copy *F57*. Although the number of tested products was small and statistical evaluation was not possible, our findings suggest a higher MAP positivity for goat products than for sheep products. These results were also consistent with the seroprevalence values at the species level at this MAP positive farm amounting to 22.8% in goats and 4.2% in sheep. According to meta-analysis including studies from the America, Asia and Europe countries and focusing on detection and molecular analysis for MAP in goat milk, the prevalence detected by MAP-specific PCR ranged from 1.9 to 37.7% [18]. These meta-analysis data showed a high degree of variability in prevalence across countries such as 7.06% (24/340) in Norway [25] to 37.74% (20/53) in India [26]. The overall prevalence of MAP DNA in milk and dairy products tested in this study fell within this percentage range (13.8%; Table 1). Positive results of our study and other worldwide studies from Scotland [27], Cyprus [28] or Italy [29] investigating the presence of MAP in cheeses are important findings. Particularly, in relation to the consumption of MAP-positive dairy products and possible susceptibility of the human organism to MAP or its association with autoimmune diseases [30]. Based on the quantification values, the number of MAP genome equivalent in goat and sheep milk was very low compared to dairy products. Since the products were made from a different batch of milk than that tested, one of the possibilities for low bacterial concentrations is the irregular excretion of MAP in the milk and/or the possible concentration of MAP cells in the final product [29]. Although all dairy products with positive MAP results were made from pasteurized milk and we only have qPCR results without information on MAP viability, it should be kept in mind that the correct pasteurization protocol according to European Union legislation must be strictly followed in order to inactivate MAP in milk. However, many other factors may also influence the survival of MAP [31,32,33].

The overall prevalence of *Staphylococcus aureus* (SA) in the samples tested in our study using qPCR was 48.8%. These measured values are consistent with the data in the literature and confirm the high incidence of SA in the milk and dairy products of small ruminants [34]. The sensitivity of conventional PCR used in our study was lower compared to qPCR systems, where the limit values were around 5 × 10^0^ of genome equivalent depending on the character of sample [35]. Based on a comparison of our results from the SA detection systems, SA qPCR positive samples with number of genome equivalent ≤10^1^ per 1 g or 1 mL were not detected as PCR positive. Therefore, it is more appropriate to use conventional PCR for testing bacterial isolates, or it can also be used for the analysis of DNA isolated from a sample with a higher concentration of target microorganisms. Based on our statistical data, we found a statistically higher chance of SA-positivity in MAP antibody-positive herds than in MAP-negative herds. The interaction between MAP and the bacterial agents of mastitis is not fully clarified, but some studies suggest that the occurrence of MAP can potentially lead to the persistence of subclinical mastitis in the herd [36]. However, staphylococcal mastitis could be only one source of SA in the samples tested in our study. Out of the 39 SA positive samples, the DNA of methicillin-resistant *S. aureus* (MRSA) was detected in 11 goat and sheep cheeses (28.2%) from five different farms. The occurrence of MRSA in sheep and goat milk is relatively common and varies primarily geographically [37,38,39,40]. The management system for small ruminants differs in some aspects compared to cattle farming. In particular, antimicrobial treatment is often an uneconomic solution for large-scale sheep and goat farmers, and slaughter is preferred. It is not, therefore, entirely clear whether the occurrence of MRSA on the five positive farms was due to inappropriate antibiotic consumption or different husbandry practices and the environment in which goats and sheep are bred. However, the fact that MRSA was not detected in the milk of the tested animals also suggests possible contamination with human host origin strains from food handlers during cheese production or processing most likely through direct hand contact with cheeses or respiratory secretions [41]. Our results of statistical analyses indicate a higher risk of SA and MRSA occurrence in tested cheese samples. The presence of MRSA in goat and sheep cheeses is a real risk for the spread of these strains in the human population, as cheese is ready-to-eat food. In addition, MRSA strains are often multiresistant (resistance to macrolides, tetracyclines, aminoglycosides and lincosamides), making treatment of infections complicated and often unsuccessful (more than 7000 deaths per year in the European Union) [42]. In our study, the *seg* and *sei* genes were detected simultaneously in one pasteurized cheese and the *sec* gene in unpasteurized cheese from another farm. Both cheeses were goat cheeses and MRSA positive. *Sec* gene is the most prevalent gene coding for classical enterotoxins detected in milk and dairy product [43]. Due to the methods used in our study, it was not possible to confirm unequivocally that the identified *seg* and *sei* genes are carried by a single strain. Nevertheless, both genes are described as part of the enterotoxin gene cluster, so this is very likely [44]. The presence of MRSA in food can also be one of the causes of the occurrence of the genes *sea*, *seb*, *sec* and *sed* encoding the enterotoxins mentioned most often in human cases of food poisoning [45]. The SE genes we revealed in goat cheeses have been previously detected in SA isolates from goats and sheep and, especially in combination with other SE genes, have been associated with human *Staphylococcus aureus* food-borne diseases (SFD) [46,47,48]. The presence of genes encoding genes for enterotoxins in cheeses does not necessarily indicate a risk of food poisoning, as it always depends on whether toxins are produced and in what quantities. Therefore, only the detection of SE directly in cheeses could confirm or refute their contribution to food poisoning. Since this was the total DNA detection in the samples including the presence of all strains of SA, it was not possible to assign these genes unequivocally to the MRSA strain, despite the obvious positivity of these samples.

Although a higher prevalence of *Listeria monocytogenes* in sheep and goat milk and cheese is reported in the literature [34], no sample in our study tested positive for the presence of *L. monocytogenes* or *Listeria* spp. However, especially immunocompromised persons should not forget that unpasteurized milk and milk products may contain other microorganisms such as *Salmonella* spp., *Campylobacter* spp., *Leptospira*, *Cryptosporidium parvum*, etc., which may pose a risk to a sensitive organism [34,49].

## 4. Materials and Methods

### 4.1. Sampling of Milk and Milk Products

The milk and dairy product samples were collected from July to November 2021 on twelve sheep and goat farms in the Czech Republic located in seven different regions. The vast majority were goat dairy farms (n = 7), followed by sheep dairy farms (n = 2), mixed sheep and goat dairy farms (n = 2) and one farm where dairy goats were kept together with a meat breed of sheep. Most of the tested farms were located in areas with a temperate climate and moderate rainfall. The average annual air temperature in these areas was around 8 °C +/− 1 °C. Depending on altitude, this annual temperature dropped proportionally to 4 °C in the foothills, where one of the tested farms was located. A cooler and wetter climate also prevailed in the next three areas with tested farms. All farms had a combined farming method with grazing. The feed of the animals was therefore based on pasture and hay ad libitum. Haylage, barley, oats, wheat, corn, sunflower, etc., root crops and mineral licks were also included, depending on the production period and the type of animals kept. The disease situation regarding caseous lymphadenitis (pseudotuberculosis) and paratuberculosis (Johne’s disease) was known on all the farms due to serological screening carried out since 2019. The majority of the tested farms did not address any other long-term health problem in the herd, with only isolated cases of mastitis caused by common pathogens such as *Staphylococcus* spp. On mixed dairy sheep and goat farms, cases of listeriosis, coccidiosis or Caprine Arthritis Encephalitis (CAE) virus were recorded. There were mortalities related to the parasite *Haemonchus contortus* on one dairy sheep farm and one mixed sheep and goat farm. The number of samples and the various types of dairy products depended on the supply of the selected farms and the size of their production. Three farms produced dairy products only for their own consumption, five farms had products available for purchase directly on the farm, and four farms both sold milk and dairy products on the farm and supplied them to the market. A total of 80 samples were collected, consisting of 18 milk samples, 28 fresh cheeses, 20 ripened cheeses and 14 yoghurts (Table 1). All dairy samples for analysis were made from pasteurized milk (except one goat cheese) and were natural without any flavorings or additives.

### 4.2. Pellet Formation and Genomic DNA Extraction

Before processing, all liquid samples (milk, yoghurt) were properly homogenized and cheese samples cut and grated. Milk samples (50 mL) were centrifuged at 4100× *g* for 45 min at 4 °C. The fat layer formed at the surface after centrifugation was removed together with almost all of the supernatant and the pellet resuspended in approximately one mL of the remaining supernatant. The resuspended pellet was transferred to a 2 mL screw-cap tube, centrifuged at 8000× *g* for 10 min and the supernatant discarded. A 15 g sample processed into a pellet was used for the analysis of cheese and yoghurt [28]. All pellets were stored at −70 °C until DNA extraction.

The Quick-DNA Fecal/Soil Microbe MiniPrep Kit (Zymo Research, Irvine, CA, USA) was used to obtain DNA. The pellets were mixed with 750 µL of Bashing Bead Buffer and ceramic beads of 0.1 and 0.5 mm in diameter. These sample tubes were mechanically homogenized for 1 min at 6400× *g* in a Magna Lyser instrument (Roche, Basel, Switzerland). Further steps were performed according to the manufacturer’s instructions. DNA was eluted into 50 µL of DNA Elution buffer.

### 4.3. qPCR

The determination of bacterial DNA in the samples was based on the detection of two specific sequences occurring independently in the genome of selected species. A generic internal amplification control (IAC) was added to each reaction at a final concentration 1 × 10^3^ to exclude false negative results. A qPCR assay targeting multiple-copy insertion sequence IS*900* was applied in all samples in the case of determination of *Mycobacterium avium* subs. *paratuberculosis* (MAP). The positivity of the qPCR was additionally confirmed and more precisely quantified by qPCR targeting *F57* [35]. The IAC nep/F57 as described by Vojkovska et al. (2015) was used in remaining bacterial qPCR systems [50]. For the determination of *Corynebacterium pseudotuberculosis* (CP), primers and hydrolysis probes targeting the *pld* gene encoding the major virulence factor toxic phospholipase D (PLD; L16587) [10,51] and the most conserved gene for pathogenic strain identification *proline iminopeptidase* (*PIP;* CP026524.1) [52] were adapted using the Primer3Plus on-line tool (https://www.bioinformatics.nl/cgi-bin/primer3plus/primer3plus.cgi (accessed on 1 January 2019). For *Staphylococcus aureus* (SA), probes and specific primers amplifying a 442 bp chromosomal DNA fragment (SA442) [53] and a part of the *nuc* gene encoding extracellular thermostable nuclease [54] were designed. Genetic determinants *23S rRNA* and *hly*III were used for detection of *Listeria* species and *Listeria monocytogenes*, respectively [55]. Detection of bacterial cells was performed on a LightCycler 480 instrument (Roche, Basel, Switzerland) following the same protocol: preincubation at 95 °C for 5 min, 47 amplification cycles (95 °C for 10 s, 60 °C for 30 s) and cooling at 40 °C. Each sample was analyzed in technical duplicate and a reaction volume of 20 µL (15 µL premix and 5 µL genomic DNA). The premix contained 2x Light Cycler^®^ 480 Probes Master (Roche) with the addition of 1 U Uracil-DNA glycosylase (Roche) and 500 nM primers at a final concentration. Probe concentrations were used depending on the optimization for each detection target [35,55]. The final probe concentrations were 200 nM for IAC and CP, 50 nM for SA442 and 100 nM for *nuc* target.

Quantification of bacterial cells was performed by the LightCycler 480 software (version 1.5.1.62, Roche Diagnostics Ltd., Rotkreuz, Switzerland) according to the calibration curve using “Fit Point Analysis”. The calibration curve was derived from a quantification plasmid standard with the plasmid construct as the starting solution. This was prepared by cloning the amplified PCR product for selected targets into the pDrive cloning vector (QIAGEN, Hilden, Germany) and transformed into chemocompetent *Escherichia coli* cells. Plasmid inserts were verified by sequencing. A standard gradient in the range of 1 × 10^5^ to 1 × 10^0^ copies/µL was prepared from the plasmid stock solution by 10-fold serial dilution in TE buffer with the addition of 50 ng/µL of Carrier DNA solution (Serva, Heidelberg, Germany). Quantification was determined for the triplex qPCR systems using a single-copy target (PLD, SA442, *hly*III). The number of genome equivalent was recalculated to one g or one mL of the original sample.

### 4.4. Conventional Multiplex PCR

SA-specific primers SA442-1 and SA442-2 derived from a 442-bp chromosomal DNA fragment [53] and primers mecA P4/mecA P7 amplifying part of the *mecA* gene found in MRSA strains [56] were selected to identify SA and methicillin-resistant *S. aureus* (MRSA). InKo1 and InKo2 universal bacterial amplification primers [57] amplifying the 241-bp product derived from highly conserved regions of the *16S rRNA* gene were used as an internal control for this PCR. The final reaction volume of 25 µL contained 2 µL of genomic DNA, PPP Master Mix (Top-Bio, Vestec, Czech Republic), 400 nM SA442, 800 nM mecA and 100 nM Inko primers. The reaction cycle consisted of 94 °C/5 min, 30 cycles of 94 °C/30 s, 59 °C/1 min, 72 °C/1 min and was terminated at 72 °C/10 min.

Detection of genes for staphylococcal enterotoxins was performed in two separate multiplex PCR assays containing primers for a different group of genes. The first reaction included specific primers amplifying the sequences of the *sed*, *see*, *seg* and *sei* genes for D, E, G and I enterotoxins, respectively [22]. The PCR was performed in 8 steps: 1. 95 °C for 10 min., 15 cycles (2. 95 °C for 1 min., 3. 64 °C for 45 s., 4. 72 °C for 1 min.) and 20 cycles (5. 95 °C for 1 min., 6. 56 °C for 45 s., 7. 72 °C for 1 min.), 8. 72 °C for 10 min. The second reaction contained toxin-specific primers for the *sea*, *seb-sec*, *sec*, *seh* and *sej* genes [22,23] to detect enterotoxins A, B, C, H and J, respectively. The PCR cycle differed from the first reaction only in the annealing temperatures, which were set to 68 °C in step 3 and 60 °C in step 6. Both PCR assays were performed in 25 µL using EliZyme^TM^ HS Robust MIX (Elisabeth Pharmacon, Brno, Czech Republic) and 300 nM primers at final concentration. The volume of template DNA was 2 µL for the first and 5 µL for the second reaction. Amplified PCR products were resolved by electrophoresis in 2% agarose gel (1x sodium borate buffer) supplemented with ethidium bromide (0.075 mg/mL) at a constant voltage of 120 V and visualized on a transilluminator. The size of the PCR products was determined using a 50 bp DNA ladder (New England BioLabs, Ipswich, MA, USA).

### 4.5. Cultivation of C. pseudotuberculosis Positive Samples

To release bacterial cells, 10 g of each CP qPCR positive cheese samples were stomached in 90 mL sodium citrate buffer (50 mM, pH 6.2) for 3 min [49]. In a 100-µL volume, homogenized samples were plated and cultivated in duplicate on brain-heart infusion (BHI) medium enriched with K₂TeO₃ (BHT-agar) at a concentration 0.125 g/L and Tween 80 [58]. The qPCR positive sheep milk sample was applied to the BHT agar plates in the same volume as the cheeses. All samples were simultaneously plated on BHI medium without added K₂TeO₃ to compare the proportion of bacteria able to grow on BHT and pure BHI medium. The inoculated agar plates were cultivated at 37 °C in a CO_2_ incubator for 48–72 h and suspected colonies were subjected to MALDI-TOF mass spectrometry. The surface of the BHT-agar plate was subsequently swabbed and the bacterial mix was resuspended in 50 µL of water for injection. These samples were lysed at 95 °C/20 min, centrifuged and used at 5 µL volume for analysis of CP by qPCR.

### 4.6. Statistical Analysis

The level of statistical significance was examined for the association between the three categories of tested samples (milk, cheese and yogurt) and the frequency of positivity for individual bacteria. The statistical significance between MAP and CP serological status of herds and the presence of bacterium SA was also investigated. Data analysis (Fisher’s exact test) was performed using package stats (R-project 4.1.2). *p*-values lower than 0.05 were considered statistically significant. Bonferroni adjustment of *p*-values was used for multiple comparison tests.

## 5. Conclusions

This study primarily applied molecular biology methods to supplement data on microbial contamination of milk and dairy products of small ruminants on Czech farms with respect to the *Corynebacterium pseudotuberculosis* and *Mycobacterium avium* subs. *paratuberculosis* serological status of sheep and goats and food safety. Although high seroprevalence values may increase the risk of possible transmission of the pathogens into the food chain, contamination of milk and milk products can be effectively prevented by following hygiene measures on farms and at all stages of the processing chain. Similarly, the greater prevalence of *Staphylococcus aureus* isolated on dairy farms from milk and subsequently from dairy products should lead to more consistent compliance with preventive measures, particularly in the case of the limit number of bacterial cells being exceeded and the presence of enterotoxigenic strains of *S. aureus* in the examined samples. In addition, the prevalence of methicillin-resistant *S. aureus* in food animals and in animal products may pose a risk of horizontal transmission of these microorganisms to humans through both direct contact with animals and consumption of contaminated food. The increasing popularity of sheep and goat products among consumers in the Czech Republic makes it necessary to monitor the disease status of dairy animals and consequently the presence and quantity of zoonotic bacteria in milk, especially if it is intended for human consumption and the production of dairy products.

## Figures and Tables

**Table 1 pathogens-11-01425-t001:** Testing sheep and goat milk and dairy products by quantitative real-time PCR (qPCR) and conventional multiplex PCR (PCR) for the presence of selected bacterial species and *Staphylococcus aureus* enterotoxin genes.

			qPCR							PCR		
Type of Sample	Origin	No.	CP		MAP		LM	SA		SA	MRSA	Exotoxins ^1^
			Pos.	GE/mL/g	Pos.	GE/mL/g	Pos.	Pos.	GE/mL/g	Pos.	Pos.	Pos.
Milk		18	1		2		0	8		4	0	0
unpasteurized	sheep	4	1	10^0^	0		0	0		nt	nt	nt
	goat	10	0		0		0	6	10^0^–10^4^	4	0	0
pasteurized	sheep	2	0		1	10^0^	0	1	10^0^	0	0	0
	goat	2	0		1	10^0^	0	1	10^0^	0	0	0
Cheese		48	6		8		0	29		21	11	2
fresh cheese		28	4		6		0	15		13	6	2
	sheep	11	3	10^0^–10^1^	2	10^1^–10^2^	0	5	10^0^	5	0	0
	goat	17	1	10^1^	4	10^1^–10^3^	0	10	10^1^–10^5^	8	6	2 ^#^
ripened cheese		20	2		2		0	14		8	5	0
	sheep	8	0		1	10^1^	0	3	10^0^–10^3^	3	3	0
	goat	12	2	10^1^	1	10^2^	0	11	10^0^–10^4^	5	2	0
Yoghurt		14	0		1		0	2		2	0	0
	sheep	8	0		1	10^1^	0	0		nt	nt	nt
	goat	6	0		0		0	2	10^0^, 10^1^	2	0	0
Total (%)		80	7 (8.8)	10^0^–10^1^	11 (13.8)	10^0^–10^3^	0	39 (48.8)	10^0^–10^5^	27 (39.7)	11 (28.2) *	2

No.—number of samples; Pos.—positive samples; nt—not tested; GE/mL/g—number of genome equivalent recalculated to 1 g or 1 mL of tested sample. CP—*Corynebacterium pseudotuberculosis*; LM—*Listeria monocytogenes*; MAP—*Mycobacterium avium* subsp. *paratuberculosis*; SA—*Staphylococcus aureus*; MRSA—Methicillin-resistant *Staphylococcus aureus.* qPCR markers: CP—PLD, *PIP*; MAP—IS*900* sensitive mainly qualitative detection, *F57* for confirmation and precise quantification; SA—SA442, *nuc*; LM—*23S rRNA*, *hly*III. ^1^ examined *S. aureus* enterotoxin genes for A, B, C, D, E, G, H, I, J enterotoxins [22,23]. ^#^ detected genes for toxin types C, G, I. * calculated from the total number of 39 *S. aureus* qPCR positive samples.

**Table 2 pathogens-11-01425-t002:** Results of serological screening by ELISA for IgG antibodies to *Mycobacterium avium* subsp. *paratuberculosis* (MAP) and *Corynebacterium pseudotuberculosis* (CP) along with detection and quantification of CP, MAP, *Staphylococcus aureus* (SA) and *Listeria monocytogenes* (LM) in milk and dairy products on small ruminant farms by quantitative real-time PCR (qPCR).

ELISA MAP/CP	No. of Farms	Farmed Animals	Type of Sample	No. of Tested Samples	CP	Amount	MAP	Amount	LM	SA	Amount
+/+	3	goats	milk	5	0		1	10^0^	0	2	10^0^, 10^4^
			cheese	14	3	10^1^ *	5	10^2^–10^3^	0	9	10^0^–10^5^
			yoghurt	1	0		0		0	1	10^0^
−/+	1	goats	milk	1	0		0		0	0	
			cheese	4	0		0		0	2	10^0^–10^1^
			yoghurt	2	0		0		0	0	
	2	sheep	milk	3	1	10^0^ *	0		0	0	
			cheese	7	3	10^0^–10^1^ *	0		0	2	10^0^
			yoghurt	3	0		0		0	0	
−/d	4	goats	milk	4	0		0		0	3	10^1^–10^2^
			cheese	8	0		0		0	6	10^1^–10^5^
			yoghurt	1	0		0		0	1	10^3^
+/d	1	goats	milk	1	0		0		0	1	10^3^
			cheese	1	0		0		0	1	10^2^
			yoghurt	1	0		0		0	1	10^2^
	1	sheep	milk	1	0		1	10^0^	0	1	10^0^
			cheese	6	0		3	10^1^–10^2^	0	6	10^1^–10^3^
			yoghurt	1	0		1	10^1^	0	0	
d/d	1	goats	milk	1	0		0		0	1	10^3^
			cheese	2	0		0		0	2	10^2^, 10^4^
			yoghurt	1	0		0		0	1	10^1^
nt/+	1	sheep	milk	2	0		1		0	0	
			cheese	6	0		0		0	0	
			yoghurt	3	0		0		0	0	

No.—number; + positive; − negative; d—dubious, positivity and negativity cannot be clearly identified; nt—not tested; Amount—number of genome equivalent in 1 mL or 1 g of tested sample. * positive CP qPCR result after examination of lysate from mixed bacterial culture grown by culturing samples on agar plates supplemented with K₂TeO₃.

## Data Availability

The original data presented in this study are included in the article. Further questions may be directed to the corresponding author.

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
