# Peer review of "First Evidence of the Presence of the Causative Agent of Caseous Lymphadenitis—Corynebacterium pseudotuberculosis in Dairy Products Produced from the Milk of Small Ruminants"

_pathogens, 2022, doi:10.3390/pathogens11121425_

Round 1

Reviewer 1 Report

This study is an identification analysis of zoonotic bacteria in milk products sampled in different regions of the Czech Republic. Molecular biology analyzes (qPCR and Multiple PCR) detect the presence of different bacterial species, with a high prevalence of Corynebacterium pseudotuberculosis. The study is interesting and could be accepted after major reviews.

-The authors have to review the paragraph of the introduction which is confusing and difficult to understand.

-The authors must insert a brief introduction on the genus Corynebacterium, on the general characteristics, pathogenicity and problems related to multi-drug resistance. Here are two examples of manuscripts that you can insert (29617406, 35625295).

-In the material and methods section, the authors do not provide information about the environmental conditions (temperature, humidity, type of climate, etc.) of the different regions of the farms. In addition, the authors should also provide information about the health of the animals, the type of diet, etc.

- In the material and methods section, “The pathological situation relating to caseous lymphadenitis (pseudotuberculosis) and paratuberculosis (Johne's disease) was known in all farms thanks to serological screening carried out since 2019”. In my opinion, the authors should have a court of control study.

-In the results section, no statistical index was calculated. As the number of samples is high, it would be advisable to carry out a thorough statistical analysis.

-In the results section, I would also advise the authors to graph analyzes of comparisons between the isolated strains in the regional differences.

- In the discussion section, authors should emphasize more comparative studies in the literature

- In the discussion section, the authors should mention the associated limits of molecular techniques used.

- I would advise the authors to write a Conclusion paragraph to summarize the salient points of the study.

Reviewer 2 Report

Dear authors,

          Overall, the data is presented well, and the article is well-written. However, the report is not focused on one aim; the title and abstract are too broad and do not reflect the findings. For example, the title and abstract state the identification of zoonotic bacteria-which are considered nonspecific statements. Additionally, this study is not novel because C. pseudotuberculosis has been detected previously in goat milk of clinical-healthy animals and animals with mastitis (ref 21). Therefore, it is not likely that animals with subclinical mastitis produce abscess that disseminates C. pseudotuberculosis into the milk.

          Furthermore, the study under review has detected other bacteria, such as M. avium subsp. paratuberculosis and S. aureus in goat milk and dairy products. These bacteria have been reported previously in several research articles. Thus, what is the value of reporting that these bacteria are found in goat milk?

Lastly, the authors did not detect L. monocytogenes in the tested samples; however, they mentioned the detection of this organism in the subtitle (2.2). 

Reviewer 3 Report

This manuscript is very well written, with very interesting results. The experimental design is well presented and conducted, the results are well presented and interesting. 

In my opinion, the improvements that can be made on the manuscript are:

(1) the authors inserted some statements on the Discussion section that need to be referenced. 

(2) The major problem of the manuscript is the Discussion section. It is very limited and need to be improved. The correlation with other findings obtained with milk samples from other European countries or even other places, not only regarding the bacteria included in the study but other bacteria should be considered.

(3) The public health implications should also be highlighted in the discussion. The zoonotic aspects of each of the pathogenic agents should be more discussed. 

Round 2

Reviewer 1 Report

The manuscript can be accepted in its current form.

Reviewer 2 Report

After addressing the reviewers' comments, the article has improved. The results and discussion flowed well in the context of the aim of the study.